# Shuttle Bus Timetable Adjustment in Response to Behind-Schedule Commuter Railway Disturbance

Yinfei Feng , Zhichao Cao * and Silin Zhang

School of Transportation and Civil Engineering, Nantong University, Nantong 226019, China
* Correspondence: caozhichao@bjtu.edu.cn; Tel.: +86-130-23562889

**Abstract:** Shuttle bus connection is a valid technique to handle unplanned problems and promote sustainable transportation. The study describes tools that facilitate the shuttle bus timetable adjustment responding to a disturbance resulting from behind-schedule trains on a commuter railway. This behind-schedule disturbance is divided in four stages allowing for different delay ranges. The problem and its solution involve different elements, such as shuttle bus route selection, stop location, and timetable adjustment. We propose a nonlinear integer programming model, in which the objective function is based on the waiting, travelling, and walking costs for passengers as well as the operation cost of the route chosen. Vehicle capacity constraints and precise passengers' waiting times are considered. A genetic algorithm and a simulated annealing algorithm combined with a priori decomposition are used to derive an efficient solution. A case study of a shuttle bus serving the Jinshan Railway in Shanghai, China, is tested to validate that, compared to the no-planning timetable, the total cost of the optimized timetable is reduced by 7.6%, especially including a dramatic reduction in the cost of passenger waiting time by 49.1%.

**Keywords:** shuttle bus timetabling; sustainable transportation; nonlinear integer programming model; shuttle bus stop locations

## 1. Introduction

The shuttle bus service is a mode of sustainable transportation for connecting users from hubs (e.g., railway stations or flight terminals) to strike the last-mile problem [1,2]. Generally speaking, the shuttle bus is responsible for accommodating passenger demands upon their exiting the schedule-based commuter railway. Matching the schedule-based (planned) timetable of the commuter railway, the shuttle bus can provide a well-connected, synchronized service with the objective of minimizing users' waiting times [3–5]. In principle, the optimal shuttle bus routes and stop locations are tailored to cover a majority of demand communities owing to their serving last-mile connecting hubs with surrounding user destinations, such as homes, offices, and other vital locations [6–8].

A commuter railway mainly helps users who live in suburbs and work in downtown areas to avoid traffic jams [9]. As long as there are neither disturbances nor disruptions, commuter railway routes and schedule-based timetables can be considered reliable with unwavering certainty; execution is precisely as planned. However, in real life operations, disturbances and disruptions are inevitable [10–12]. Cacchiani et al. [13] elaborate on two distinctive differences between the characteristics of "disturbances" and "disruptions". Disturbances, which are formally defined as the malfunctions, mistakes, or deviating states occurring within the infrastructure or operating conditions, are common. In this study, we focus specifically on the shuttle bus timetable adjustment in response to a behind-schedule commuter railway disturbance.

The behind-schedule phenomenon resulting in disturbances imposes unanticipated congestion from accumulated passengers incessantly exiting commuter railway stations. Likewise, shuttle bus services are deprived of the opportunity of feeding into well-connected

and timely transfers. Therefore, timely adjustment of the shuttle bus schedule necessarily entails responding to behind-schedule commuter railway trains [14]; however, this involves taking the uncertainty of the nature of the disturbance into account. The shuttle bus is inclined, by rational programming, to propose multiple dimensional adjustments including route selections, stop locations, and timetable in order to handle different delay ranges.

To date, although there are extensive railway rescheduling approaches, there is little in the literature that pays ample attention to the follow-up feeder service adjustment that is closely connected with the foregoing reschedule. It refers to the shifts in both departure times and headways of the shuttle bus in our study. The defined problems in the paper cover the threefold aspects as follows:

1. An integrated challenge with respect to bus stop locations, optional routes, and optimized timetable;
2. The elimination process of different delay ranges predefined as four stages, which require different responses/shifts in individual stages;
3. A nonlinear integer programming model is proposed, in which the objective function takes passengers' waiting, travelling, and walking costs into account as well as combining them with the operation cost of the entire route. A genetic algorithm and simulated annealing are used to encode the candidate bus stops by searching the traversal of bus route options. A well-adjusted timetable and bus routes showing stop locations are generated by bridging the connection gap derived from the behind-schedule commuter railway.

### 1.1. Literature Review

In what follows, we introduce shuttle bus optimization and, particularly, its application accommodating the commuter railway as per various strategies or tactics. The existing literature provide a basis for the study and contributes to a better understanding of aspects of research addressed by our study.

Jerby and Ceder [15] proposed a route design approach for a shuttle bus system, in which the potential passenger demand for optional routes is estimated quantitatively. The study aims at maximizing the number of passengers and minimizing walking distances. Lownes and Machemehl [16] proposed a mixed integer model for a single-route circulation programming problem that was solved by a tabu search for large scale computation. Three solution approaches, i.e., enumeration, One Tree, and tabu search, are applied for the small and large networks. Yu et al. [17] proposed a bilevel nonlinear mixed integer programming model to minimize the total tour cost of bus passengers and operators. They considered passengers' walking times and developed a tabu search with a local search strategy and neighborhood evaluation tool but did not take into account passengers' waiting time. Jin et al. [18] introduced a bus bridging tactic that takes commuter demand during a disruption horizon into consideration. They generated demand-responsive bus route candidates via a column generation and established a path-based multicommodity network flow model. The aim is to identify the suitable bus routes and fleet size assignment. Kong et al. [19] presented a shuttle bus route formulating method by means of crowd sourced mobile data. A dynamic programming algorithm was used to obtain the optimal routes of shuttle buses. Cao et al. [20] pursued the optimized fleet size by striking the tradeoff between timetabling and vehicle scheduling by using the skip-stop strategy. A binary variable iteration heuristic is devised to deal with the large-scale case. Cao and Ceder [21] proposed a real-time timetable adjustment model as per holding and speed-changing strategies to attain reliability. Liang et al. [22] designed a robust bus bridging management plan as a remedial reaction to a rail transit disruption. They developed a path-based multi-commodity flow formulation allowing for running time uncertainty of bus bridging. The model was resolved by a column generation algorithm. Cao et al. [23] developed an approach for deriving an optimized timetable for multiphase demands featuring three operation tactics: marshaling, skip-stop, and robust adjustment. Wang et al. [24] concentrated on a shuttle bus routing optimization problem under urban rail transit emergencies accounting for evac-

uation priorities. The approach is beneficial for both railway disruptions and interruptions. Cao et al. [25] proposed a decision-making, coupling–decoupling, strategic programming model to achieve capacity dynamic optimization. Cao et al. [26] considered passengers' service-frequency satisfaction in terms of waiting times, passengers' perception of riding comfort via seat availability, and planned passenger load ratio linked with operation efficiency. An estimation framework for bus timetabling is built for the planning stage. Wu et al. [27] proposed a mixed-integer linear programming model as per shared ride and shuttle bus deployment. A modified Lagrangian relaxation algorithm and rolling horizon scheme were proposed to attain solution efficiency in solving a large-scale case. They could get better results by using exact algorithms.

Unlike the existing studies mentioned which resolved certain unanticipated situations by deploying temporary bus bridging, this study seeks timely adjustment on daily/routine shuttle bus systems specifically responding to behind-schedule situations that commuter railways encounter. In other words, on rescheduling shuttle bus timetables there is a lack of attention to newly-rescheduled commuter railways providing connection services. Table 1 illustrates the features and merits of the existing literature by comparing these closely related and timely studies.

**Table 1.** Comparison of relevant studies on shuttle bus service and its application on the commuter railway.

| Publication (Chronologically) | Motivation from Disturbance/Disruption | Capacity Constraint | Considers Waiting Times | Objective(s) | Solution Algorithms | Case |
|---|---|---|---|---|---|---|
| Jerby and Ceder (2006) [15] | No | No | No | To maximize the demand potential of the route links | A heuristic algorithm based on the first option | Experiments |
| Lownes and Machemehl (2010) [16] | No | No | No | To minimize the total cost of operating bus, in-vehicle travel time, unserved demand and walking | Tabu search | Experiments |
| Yu et al. (2015) [17] | No | No | No | To minimize the total tour cost for passengers and operators with respect to minimization in passenger walking time | Tabu search | Experiments and real |
| Jin et al. (2016) [18] | Disruptions | Yes | Yes, as a sub-problem | To minimize the total increase in journey time of all commuter groups over the entire disrupted period | A column generation procedure | Experiments |
| Kong et al. (2018) [19] | No | Yes | No | To minimize the operating distance | A dynamic programming algorithm | Real |
| Cao and Ceder (2019) [21] | No | Yes | Yes | To minimize passengers' in-vehicle times, waiting times and fleet sizes | A binary variable iteration method | Experiments |
| Liang et al. (2019) [22] | Disruptions | Yes | No | To minimize passengers' travel costs on railway, in-vehicle costs, transfer costs and bus bridging operation costs | A column generation procedure | Experiments |
| Wang et al. (2021) [24] | Disturbances and disruptions | No | No | To minimize the total fixed edge cost based on the OD paths | A two-step method | Experiments |
| Wu et al. (2022) [27] | No | Yes | Yes | To minimize passengers' waiting times, in-vehicle times, and operation costs | Lagrangian relaxation and rolling horizon optimization | Experiments and real |
| This paper (2022) | Disturbances | Yes | Yes | To minimize passengers' waiting, travelling and walking costs as well as operation costs for the entire route | Genetic algorithm and simulated annealing | Experiments and real |

### 1.2. Contributions

In facilitating shuttle bus development, it is neither conventional nor trivial to allow for route selection, stop locations, and timetable adjustment jointly so as to respond to left-behind needs with respect to commuter railway schedule disturbances. Thus, the contribution of our study is a new integrated shuttle bus programming model with:

1. The joint optimization between tactic level routes, stops, and strategy level timetabling;
2. A nonlinear integer programming model proposed to deal with the left-behind schedule adjustment problem with the objective of minimizing net supply and demand interests with regard to walking, waiting, and traveling times;
3. Two algorithms, i.e., a genetic algorithm and simulated annealing adopted to facilitate the solution efficiency.

### 1.3. Layout

The layout of this paper in the following sections is as follows: Section 2 presents the problem statement, assumptions, and nomenclature. Section 3 proposes a nonlinear mixed integer programming model for the shuttle bus's route planning and timetabling problem. The scale of commuter railway delays and the corresponding subprocesses are pre-defined. Section 4 decomposes the model and gets the optimal stop location set in which the optimized shuttle bus service routes and the optimized timetables in each subprocess are derived accordingly by using a genetic algorithm and simulated annealing. Section 5 tests the case of the Jinshan Railway in Shanghai, China, and conducts a sensitivity analysis. Section 6 presents the conclusions and suggestions for further research.

## 2. Problem Statement

### 2.1. Study Context

Figure 1 represents the geographical topology of the shuttle bus network serving the commuter railway. Stop location candidates (black points) surrounding the destination centroids (orange squares) allow us to generate the optional routes, depicted by the black lines of Figure 1. The demand origin, namely, the railway station, is marked by blue diamond shapes. Typically, the cost–benefit estimation drives us to capture/elect the optimized routes from the candidate set.

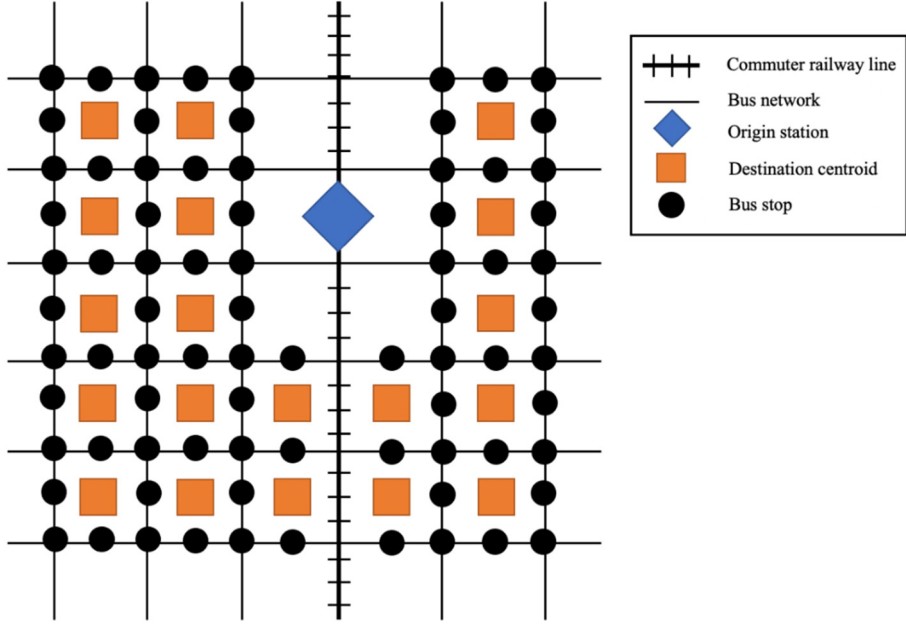

**Figure 1.** Example of the shuttle bus service.

In this sense, it suggests a sophisticated route necessarily covers the overall destination centroids, well aware of how costs are saved via our modelling.

In the real world, matters such as congested demand, weather conditions, or temporary power loss and so on, have always subjected railway systems to unanticipated disturbances. Given these unforeseen events, the disturbance is inclined to trigger departure delays and leads to unscheduled headway between the disturbance train and the predecessor/successor ones. In that case, in order to facilitate transporting the passenger accumulation in an efficient manner, the shuttle bus should respond in a timely enough manner to enable and provide a reasonable adjustment for meeting these unanticipated demands.

Figure 2a shows the different stages of the commuter railway once encountering a disturbance with a consequence ranging from "normal" to "abnormal" transition, and subsequently recovering to "normal". Owing to the shuttle bus loop, the origin and terminal stops of the shuttle bus route are located at a railway station which we define as the *origin station*, shown in Figure 2. *Origin station* also refers to the blue diamond shape in Figure 1. Figure 2b demonstrates the individual processes of shuttle buses that are triggered to adjust correspondingly, yielding to a commuter railway behind schedule. "Farthest bus stop" refers to the bus stop at the greatest distance from the *origin station*. Furthermore, as one example of shuttle bus timetable adjustment in Figure 2b, taking multi-scales of delays into consideration, we classified the different headway intervals with comparison to the planned headway. It should be noted that the headway time-intervals are yielding to their own separate periods as defined in Figure 2a.

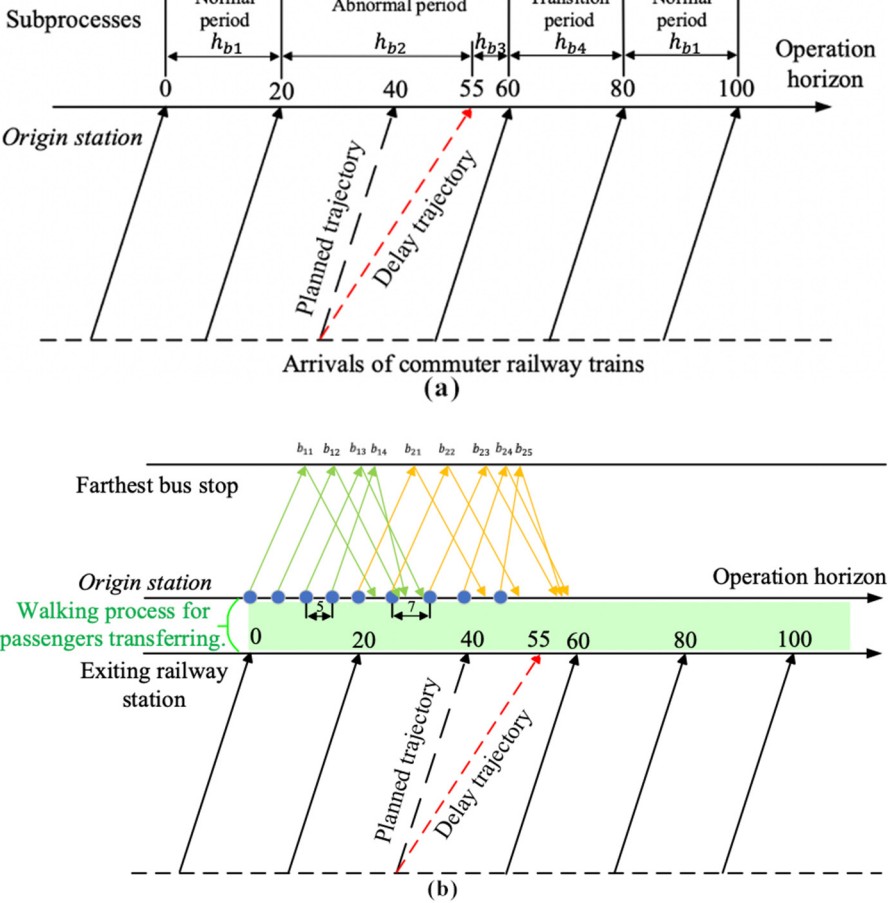

**Figure 2.** Demonstration of shuttle bus serving commuter railway during different periods: (**a**) individual stages of the commuter railway undergoing the disturbance; (**b**) headway interaction between shuttle bus and commuter railway allowing for delays.

For the sake of clear representation, the bus headways of four sub-processes are denoted by $h_{b1}$ to $h_{b4}$, respectively. Four headways correspond to four stages $s_1$ to $s_4$. $s_1$ is the normal operation horizon in which commuter railway trains and shuttle buses precisely execute the scheduled timetables/headways as planned. $s_2$ and $s_3$ are behind-schedule situations. The headway period between the delayed train $i$ and its predecessor as well as the successor of train $i$ on a given line are denoted as $s_2$ and $s_3$, respectively. In other words, $s_2$ explicitly refers to the period between the delayed train and the predecessor train. Thus, the headway is more than that of the normal schedule $s_1$. Accordingly, the headway of the shuttle bus system necessarily entails a longer headway than normal. $s_3$ implies the headway between the delayed train and successor train, which leads a shorter headway than that of the initial plan/normal. Correspondingly, the headway of the shuttle bus warrants reducing. $s_4$ is in the transition period, which means that the headway of the commuter railway returns to normal. Nevertheless, the headways of the shuttle buses may still be larger than those before disturbances.

## 2.2. Assumption

The aforesaid problem statement motivates us to propose the following assumptions:

1.　The delays caused by disturbance have the capability of recovering to a normal/plan by rescheduling techniques along the railway operation horizon;
2.　A great number of passengers alighting from trains have a fixed number of gates to exit and follow the first-come first-served principle (FCFS). The OD (origin destination) proportion of passenger demand arriving at the *origin station* is stable;
3.　Rigorous load capacity is fixed owing to the homogeneous vehicle type. In other words, the passengers who exceed the upper load of capacity threshold cannot board vehicles. In addition, overtaking is allowed;
4.　The destinations desired by passengers are located approximately in the center of the defined districts. Destination information for these commuter passengers is known to assure exact destination districts;
5.　Traffic accidents and vehicle breakdown do not occur on the horizon of the study;
6.　A shuttle bus is featured by a single-directional loop route. Notice that shuttle buses do not pick up the passengers along the route, rather they are customized to accommodate commuter railway demand from the *origin station*. It means that there are only *alighting passengers* as long as the shuttle buses depart from the *origin station*;
7.　The number of trips is judged by an evaluation metric of operation cost with a hypothesis of sufficient fleet sizes.

Herein, the motivations of the aforementioned assumptions are presented in detail. Assumption (1) refers to a controllable scale of delay that conforms to the definition of disturbance [13]. Thus, the knock-on effect of continuing delay is not taken into consideration. Then, Assumption (2) specifies an even arrival rate of passengers, the rule of which fits FCFS owing to the exiting queue. Commuter passengers are mainly fixed and thereby their OD are steady, which facilitates collecting their travel information. Next, Assumption (3) stipulates strict capacity constraint and bunching elimination. Assumption (4) achieves valid service areas covered by shuttle bus, and Assumption (5) guarantees an available study context. Finally, Assumption (6) and Assumption (7) clarify a unidirectional/one-way service style and assess the number of trips instead of the fleet size as one objective of the model, respectively.

## 2.3. Nomenclature

A shuttle bus system is composed of stop set $I = \{i, j, k\}$ and destination centroid set $G = \{g\}$. A given set of trips $B = \{b_{sn}\}$ is responsible for the connection service. The set of processes undergoing disturbance is denoted by $S = \{s\}$.

Given that the OD matrix is captured, the number of passengers $d_g$ who are desired for their destinations are known as input. Yielding to Assumption (5), the running time candidates $t_{ij}$ between stops $i$ and $j$ are fixed and obtained. Walking link is depicted by

$y_{ig}$ for seeking the distance/time $\tau_{ig}$ between the stop location and destination centroid as estimation criteria. $d_b$ means the number of total trips, which are equal to the number of elements in set $B$.

To be clear, the notations and parameters used throughout the paper are explained in Table 2.

**Table 2.** List of notations and parameters along with explanatory descriptions.

| | Sets |
|---|---|
| $G$ | Set of destination centroids, $G = \{g\}$ |
| $I$ | Set of bus stops, $I = \{i, j, k\}$ |
| $B$ | Set of trips, $B = \{b_{sn}\}$, which indicates that trip $n$ belongs to process $s$ |
| $S$ | Set of subprocesses, $S = \{s\}$ |
| $U$ | Set of users, $U = \{u\}$ |
| | **Parameters** |
| $d_g$ | Number of passengers who are willing to reach destination centroid $g$ (unit: pax) |
| $t_{ij}$ | Bus travel time between bus stops $i$ to $j$ (unit: min) |
| $\tau_{ig}$ | Walking time of passengers from stop $i$ to destination centroid $g$ (unit: min) |
| $c_o$ | Vehicle operation cost per unit time (unit: USD /min) |
| $c_{iv}$ | In-vehicle travel cost for each passenger per unit time (unit: USD /min) |
| $c_{ov}$ | Out-of-vehicle walking cost for each passenger per unit time (unit: USD /min) |
| $c_{wv}$ | Waiting cost for each passenger per unit time (unit: USD /min) |
| $C_b$ | Capacity of vehicle $b$ (unit: pax/vehicle) that is fixed subject to Assumption (3) |
| $\lambda$ | Passenger arrival rate (unit: pax/min) |
| $\delta$ | Passenger drop-on/drop-off time (unit: sec) |
| $\delta_0$ | Opening/closing door time (unit: sec) |
| $t_u$ | Time slice when passenger $u$ arrives at *origin station* |
| | **Variables** |
| $d_b$ | Number of total trips |
| $\delta_i$ | Total alighting time of passengers at bus stop $i$ over all trips (unit: min) |
| $t_{b_{sn}}$ | Departure time of trip $n$ in subprocess $s$ |
| $w_{ij}$ | Total number of passengers assigned at the link between stops $i$ and $j$ over all trips (unit: pax) |
| $wt_u$ | Waiting time for passenger $u$ at *origin station* (unit: min) |
| | **Decision variables** |
| $z_i$ | Bus stop selection indicator, which specifies that bus stop $i$ is elected if $z_i = 1$; otherwise $z_i = 0$ |
| $x_{ij}$ | Bus link selection indicator, which implies that one bus link candidate between bus stops $i$ and $j$ is elected if $x_{ij} = 1$; otherwise $x_{ij} = 0$ |
| $y_{ig}$ | Passenger link selection indicator, which suggests there are passengers who choose the walk link from stop $i$ to destination centroid $g$ (i.e., alighting from bus stop $i$ and desired to destination centroid $g$) if $y_{ig} = 1$; otherwise $y_{ig} = 0$ |
| $e^u_{b_{sn}}$ | Passenger boarding selection indicator, which denotes that the passenger $u$ boards the trip $n$ in subprocess $s$ from *origin station* if $e^u_{b_{sn}} = 1$; otherwise $e^u_{b_{sn}} = 0$ |
| $h_{bs}$ | Headway of shuttle bus in subprocess $s$ (unit: min) |

## 3. Modelling Formulation

### 3.1. Objective Functions

This section introduces objective functions of the model in detail. Objective function (1) is mainly involved with passengers' and operators' interests, which is to minimize four components, i.e., passengers' costs for (i) walking, (ii) travelling, (iii) waiting, and (iv) operation cost.

$$\min c_{ov} \sum_{i \in I} \sum_{g \in G} \tau_{ig} d_g y_{ig} + c_{iv} \left( \sum_{i \in I} \sum_{j \in I \setminus \{i\}} t_{ij} w_{ij} + \sum_{i \in I} \sum_{j \in I \setminus \{i\}} \delta_i w_{ij} \right) + c_{wv} \sum_{u \in U} wt_u + c_o \left( d_b \sum_{i \in I} \sum_{j \in I \setminus \{i\}} t_{ij} x_{ij} + \sum_{i \in I} \delta_i \right). \quad (1)$$

In the objective function, $c_{ov}$, $c_{iv}$, $c_{wv}$, and $c_o$ are monetary coefficients, respectively. The Objective function (1) minimizes the net travel benefits given by (i) walking times from stop to destination centroid; (ii) in-vehicle travelling times for passengers; (iii) waiting times at *origin station*; and (iv) shuttle buses' travelling times. These time costs are assessed based on an identical monetary unit, i.e., U.S. dollars.

Next, Section 3.2 will state the constraints of the shuttle bus.

### 3.2. Shuttle Bus Constraints

Constraints (2) and (3) specify link $x_{ij}$ is generated once stops $z_i$ and $z_j$ are elected, which are equal to 1. Constraint (4) validates the number of passengers $w_{ij}$ distributed to the link $x_{ij}$ only when link $x_{ij}$ is selected (equal to 1). Constraint (5) achieves an equilibrium assignment, in which the number of passengers $d_g$ aiming for their destination $g$, $d_g y_{ig}$, is equal to in-vehicle passengers $w_{ki}$ minus the ones $w_{ij}$ alighting at stop $i$ already. Constraint (6) clarifies that the number of passengers alighting at all stops along the route (right side) is equal to the sum of ones boarding from the *origin station* $i_0$ (left side). Constraint (7) ensures that all passengers have alighted already once the shuttle bus arrives at terminal stop (i.e., back at the *origin station* $i_0$). Constraint (8) assures that the capacity summation of total shuttle buses $C_b d_b$ is more than total passenger demand $\sum_{g \in G} d_g$. Constraint (9) presents the number of passengers boarding one single bus as less than the vehicle capacity $C_b$. Constraint (10) stipulates the minimum and maximum headways of the shuttle bus system. Especially, the headway cannot exceed the duration of each subprocess. Constraint (11) states that the bus can depart from the *origin station* only if meeting the headway of the subprocess. Constraint (12) calculates the waiting time of passengers at the *origin station*, which depends on the time-window between the arrival time at *origin station* and the boarding time. Constraint (13) ensures that the waiting time of passenger $u$ is non-negative, i.e., if passengers want to get on the bus, they must arrive at *origin station* earlier than the departure time of trip $n$ at *origin station*. Constraint (14) expresses the relationship between the bus dwell time and the number of alighting passengers, as well as door opening and closing times. Constraint (15) validates that the passenger flow of each link is non-negative. Constraint (16) manifests the relationship between bus stop selection $z_i$ and passenger walking links $y_{ig}$. Only when the bus stop $i$ is selected is the corresponding walk link valid. Constraint (17) strictly imposes one destination centroid corresponding to one stop. Constraints (18)–(21) are decision variable constraints.

$$\sum_{j \in I \setminus \{i\}} x_{ij} = z_i \quad \forall i \in I, \tag{2}$$

$$\sum_{i \in I \setminus \{j\}} x_{ij} = z_j \quad \forall j \in I, \tag{3}$$

$$w_{ij} \leq \left( \sum_{g \in G} d_g \right) x_{ij} \quad \forall i, j \in I, \tag{4}$$

$$\sum_{k \in I \setminus \{i\}} w_{ki} - \sum_{j \in I \setminus \{i\}} w_{ij} = \sum_{g \in G} d_g y_{ig} \quad \forall i \in I, \tag{5}$$

$$\sum_{i \in I \setminus \{i_0\}} w_{i_0 i} = \sum_{i \in I \setminus \{i_0\}} \sum_{g \in G} d_g y_{ig}, \tag{6}$$

$$\sum_{k \in I \setminus \{i_0\}} w_{ki_0} = 0, \tag{7}$$

$$C_b d_b \geq \sum_{g \in G} d_g, \tag{8}$$

$$\sum_{u \in U} e^u_{b_{sn}} \leq C_b \quad \forall b_{sn} \in B, \tag{9}$$

$$h_{min} \le h_{bs} \le h_{max} \quad \forall s \in S, \tag{10}$$

$$t_{b_{sn}} - \sum_{t=1}^{s-1} h_{ct} - \left\lfloor \frac{t_{b_{sn}} - \sum_{t=1}^{s-1} h_{ct}}{h_{bs}} \right\rfloor h_{bs} = 0 \quad \forall b_{sn} \in B, \tag{11}$$

$$wt_u = \left( \sum_{b_{sn} \in B} e^u_{b_{sn}} t_{b_{sn}} \right) - t_u \quad \forall u \in U, \tag{12}$$

$$e^u_{b_{sn}} t_u \le t_{b_{sn}} \quad \forall u \in U \quad \forall b_{sn} \in B, \tag{13}$$

$$\delta_i = \begin{cases} \delta \sum_{g \in G} y_{ig} d_g + d_b \delta_0 & \forall y_{ig} = 1 \\ 0 & \forall y_{ig} = 0 \end{cases} \quad \forall i \in I, \tag{14}$$

$$w_{ij} \ge 0 \quad \forall i, j \in I, \tag{15}$$

$$y_{ig} \le z_i \quad \forall i \in I, \tag{16}$$

$$\sum_{i \in I} y_{ig} = 1 \quad \forall g \in G, \tag{17}$$

$$z_i \in \{0, 1\} \quad \forall i \in I, \tag{18}$$

$$x_{ij} \in \{0, 1\} \quad \forall i, j \in I, \tag{19}$$

$$y_{ig} \in \{0, 1\} \quad \forall i \in I, \tag{20}$$

$$e^u_{b_{sn}} \in \{0, 1\} \quad \forall u \in U. \tag{21}$$

## 4. Solution Algorithm

### 4.1. Decomposition

To solve this problem, we execute the decomposition preprocess, i.e., pre-considering the shortest distance from stops to destination centroids. Afterwards, obtained from presolved outcomes, the new objective function links the selected bus stops to design a route and then generates the shuttle bus timetable seeking the minimum cost.

$$\min c_{iv} \left( \sum_{i \in I} \sum_{j \in I \setminus \{i\}} t_{ij} w_{ij} + \sum_{i \in I} \sum_{j \in I \setminus \{i\}} \hat{\delta}_i w_{ij} \right) + c_{wv} \sum_{u \in U} wt_u + c_o \left( d_b \sum_{i \in I} \sum_{j \in I \setminus \{i\}} t_{ij} x_{ij} + \sum_{i \in I} \hat{\delta}_i \right), \tag{22}$$

$$\sum_{j \in I \setminus \{i\}} x_{ij} = \hat{z}_i \quad \forall i \in I, \tag{23}$$

$$\sum_{i \in I \setminus \{j\}} x_{ij} = \hat{z}_j \quad \forall j \in I, \tag{24}$$

$$w_{ij} \le \left( \sum_{g \in G} d_g \right) x_{ij} \quad \forall i, j \in I, \tag{25}$$

$$\sum_{k \in I \setminus \{i\}} w_{ki} - \sum_{j \in I \setminus \{i\}} w_{ij} = \sum_{g \in G} d_g \hat{y}_{ig} \quad \forall i \in I, \tag{26}$$

$$\sum_{i \in I \setminus \{i_0\}} w_{i_0 i} = \sum_{i \in I \setminus \{i_0\}} \sum_{g \in G} d_g \hat{y}_{ig}, \tag{27}$$

$$\sum_{k \in I \setminus \{i_0\}} w_{k i_0} = 0, \tag{28}$$

$$C_b d_b \ge \sum_{g \in G} d_g, \tag{29}$$

$$\sum_{u \in U} e^u_{b_{sn}} \le C_b \quad \forall b_{sn} \in B, \tag{30}$$

$$h_{min} \le h_{bs} \le h_{max} \quad \forall s \in S, \tag{31}$$

$$t_{b_{sn}} - \sum_{t=1}^{s-1} h_{ct} - \left\lfloor \frac{t_{b_{sn}} - \sum_{t=1}^{s-1} h_{ct}}{h_{bs}} \right\rfloor h_{bs} = 0 \quad \forall b_{sn} \in B, \tag{32}$$

$$wt_u = \left( \sum_{b_{sn} \in B} e_{b_{sn}}^u t_{b_{sn}} \right) - t_u \quad \forall u \in U, \tag{33}$$

$$e_{b_{sn}}^u t_u \le t_{b_{sn}} \quad \forall u \in U \quad \forall b_{sn} \in B, \tag{34}$$

$$w_{ij} \ge 0 \quad \forall i, j \in I, \tag{35}$$

$$x_{ij} \in \{0,1\} \quad \forall i, j \in I, \tag{36}$$

$$e_{b_{sn}}^u \in \{0,1\} \quad \forall u \in U. \tag{37}$$

Herein, we rewrite Constraints (1)–(21) to derive new Constraints (22)–(37). In brief, we eliminate the decision variables in determining the optional bus stop locations and computing passenger walking time. Thereby, we can reduce the traversal on route selection. $\hat{\delta}_i$ refers to alighting time of passengers at optimized stop candidate $i$ in Constraint (22). $\hat{z}_i$ and $\hat{z}_j$ denote bus stops $i$ and $j$ selected in the pre-solved sub-problem in Constraints (23)–(24). $\hat{y}_{ig}$ is the optimized passenger link in the pre-solved sub-problem in Constraints (26)–(27).

Objective function (38) is created for the purpose of pre-determining the shuttle bus routes and finding the optimized route as per genetic algorithm and simulated annealing. Explicitly, given the OD matrix of passenger demand, the aim is to minimize the total cost of one refined trip as per the testing of different routes.

$$\min \sum_{i \in I} \sum_{j \in I \setminus \{i\}} t_{ij} w_{ij} + \sum_{i \in I} \sum_{j \in I \setminus \{i\}} \hat{\delta}_i w_{ij}. \tag{38}$$

Objective function (38) addresses the travel time and alighting times of passengers that refer to the first and second terms of Objective function (22), respectively. Initially, Objective function (38) helps for the shortest (precisely, that means minimum cost) path problem of TSP.

### 4.2. Genetic Algorithm

We adopt a genetic algorithm (GA) [28,29] to resolve the nonlinear integer programming model proposed, rather than employ the TSP approach because TSP is incapable of adjusting the shuttle bus headway to handle the behind-schedule commuter railway. The OD matrix of passenger demand is known based on real-time information collection and historical data. GA is applied twice to seek the shuttle bus service's optimized routes and timetables, respectively. The procedures of addressing this problem are presented in detail below.

#### 4.2.1. Genetic Algorithm for the Routing Problem

First, GA is used to solve the routing problem. The detailed steps are introduced.
Step 1: Initialization.
Generate the evolution time counter $t = 0$, stipulate the maximum evolution times $T$, and define random individuals $M$ in the initial population $P(0)$ with the generation of the initial route candidate $r$ as the initial population.
Step 2: Personal evaluation.
Calculate the fitness of everyone in the population $P(t)$. For the purpose of optimizing routes, different route candidates are tested with the objective of minimizing total passengers in-vehicle travel costs.
Step 3: Select operation.
Apply the selection operation to the population. The aim of the selection is for the next generation to inherit the optimized individuals, or alternatively to generate new

individuals through combining and crossover. Then, they should be inherited by the next generation. The selection operation is the basis of the fitness evaluation of the individuals in the population. We use the roulette selection, which is a replay random sampling method. The probabilities of each individual accessing the next generation is equal to the inverse proportion of its fitness value to the total fitness of the total population. A smaller fitness value poses a higher possibility for seeking a superior shuttle bus route.

Step 4: Crossover operation.

Exert the crossover operation on the population. Crossover relates to the activities of replacing and recombining part of the formation of the two parent individuals to result in a new individual. Suppose there are G bus stops. In the crossover operation process, $\left\lfloor \frac{G}{2} \right\rfloor$ stops of the individual P1 is randomly selected. If this part is selected, all the genes of this part are exchanged with the corresponding genes of individual P2. Figure 3 shows the crossover operation process of individual P1 and P2 (the omitted part also has exchange). Numbers of P1 and P2 represent bus stop indices. Crossover operation ensures that the resulting offspring individual is a feasible solution. By crossing operations, new routes are generated.

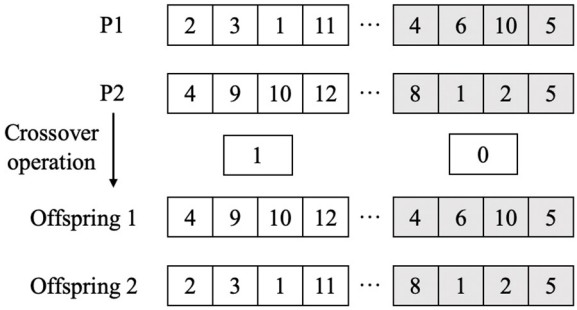

**Figure 3.** Crossover operation on routes.

Step 5: Mutation operation.

Implement mutation operator for the population; that is, the gene value on some loci of the individual string in the population is changed. As shown in Figure 4, a mutation mask is randomly generated before performing the mutation operation. For the selected individual's bus stop part, the mutation operation is performed on the $\left\lfloor \frac{G}{2} \right\rfloor$ genes.

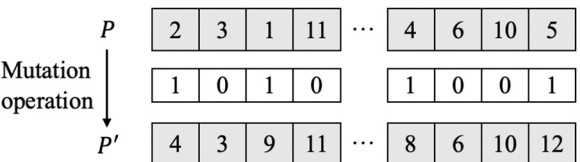

**Figure 4.** Mutation operation on routes.

Step 6: Termination.

Until $t = T$, the individual within the maximized fitness derived from the evolution is utilized as the optimized outcoming output and the traversal is brought to an end.

By means of the foregoing six steps, after inputting the initial data (i.e., optimal bus stops set and basic passenger information), an optimized route $r'$ is obtained.

### 4.2.2. Genetic Algorithm for the Timetabling Problem

In addition, GA is used to solve the timetabling problem. The detailed steps are introduced.

Step 1: Initialization.

Generate the evolution time counter $t = 0$, stipulate the maximum evolution times $T$, and define random individuals $M$ in the initial population $P(0)$. In generating the initial timetable, the random bus headways $h_{bs}$ in each subprocess $s$ is defined as the initial population.

Step 2: Personal evaluation.

Calculate the fitness of everyone in the population $P(t)$. For the purpose of optimizing timetable, the headway of shuttle bus service in each process $s$ is tested with the objective of minimizing total passengers traveling, waiting, and vehicle operation costs.

Step 3–Step 6 are similar to the steps in Section 4.2.1

By means of the foregoing six steps, after inputting the initial data (i.e., optimized route and basic passenger information), the optimized shuttle bus headway $h'_{bs}$ is derived.

The Algorithm 1 flowchart reflecting the details is shown in Figure 5. The following pseudocode presents the algorithmic procedures by which the shuttle bus timetable candidates are derived.

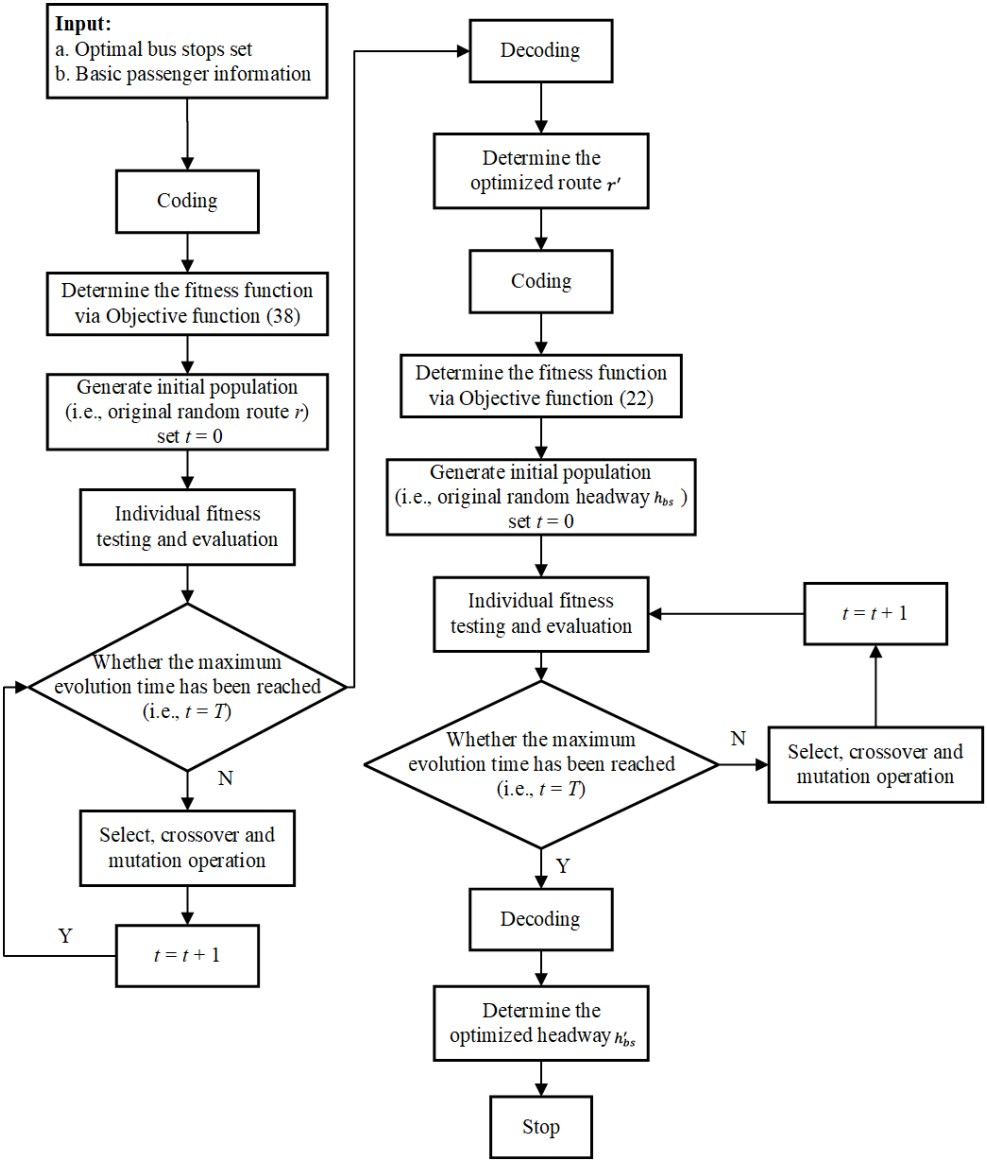

**Figure 5.** Flowchart of decomposition-based genetic algorithm.

---

**Algorithm 1.** Procedures for generating the shuttle bus route/timetable by GA

---

**input** the initial populations containing the route candidate/ the headway of each subprocess
*Pop*, terminate condition *T*, evolutionary population generation *t*
**output** *New_Pop*
**initialize** $t \leftarrow 0$, fitness function value $\leftarrow +\infty$
fitnessP (*Pop*)
**while** $(t \leq T)$ **do**
  GA-operationP (*Pop*)
fitnessP (*Pop*)
  *New_Pop* $\leftarrow$ *Pop*
  *t++*
**end while**
**return** *New_Pop*

---

### 4.3. Simulated Annealing

Compared to GA, simulated annealing (SA) [30,31] has a global search superiority by which fewer results are trapped in the local optimum. SA is derived from the annealing process of solid materials in physics starting with an initial temperature and followed by a temperature decrease. The global optimal solution is found randomly in the solution space, so it belongs to a probability-based search scope. Similar to GA, this paper applies SA twice to seek the shuttle bus service's optimized routes and timetables, respectively. The procedures for addressing this problem are presented in detail as follows.

#### 4.3.1. Simulated Annealing for the Routing Problem

First, SA is used to solve the routing problem in a similar way.

Step 1: Initialize, choose an initial solution, i.e., the route candidate $x_0$ to calculate fitness $z(x_0)$, i.e., total passengers in-vehicle travel costs, let $x_{best} = x = x_0$, use the given starting and termination temperatures $T_0$ and $T_f$, respectively, and set iteration $k = 0$.

Step 2: Randomly choose a solution $x_k$ in the neighborhood and calculate the objective function increment $\Delta f = f(x_k) - f(x)$. For neighbor solution generation, the order of any two bus stops can be exchanged or the elements between the two bus stops can be rearranged in reverse order.

If $\Delta f < 0$, then let $x = x_k$. Otherwise, generate a random number $\xi = U(0,1)$. If the random number is smaller than the transfer probability $P(\Delta f, T)$, then let $x = x_k$.

Step 3: Decrease the temperature $T$.

Step 4: If the maximum number of iterations $k_{max}$ or the minimum temperature $T_f$ is reached, then stop. Let $x_{best} = x$, or else go to Step 2.

#### 4.3.2. Simulated Annealing for the Timetabling Problem

In addition, SA is used to solve the timetabling problem, whose steps are summarized as follows.

Step 1: Initialize, choose an initial solution $x_0$, i.e., the initial timetable, i.e., the random bus headways $h_{bs}$ in each subprocess $s$, calculate fitness $z(x_0)$, i.e., total passengers traveling, waiting, and vehicle operation costs, let $x_{best} = x = x_0$, use the given starting and termination temperatures $T_0$ and $T_f$, respectively, and set iteration $k = 0$.

Step 2–Step 4 are similar to the steps in Section 4.3.1

The following pseudocode presents the algorithmic procedures by which the shuttle bus timetable candidates are derived (Algorithm 2).

---

**Algorithm 2.** Procedures for generating the shuttle bus route/timetable by SA

---

    **input** the initial solution $x_0$, termination temperature $T_f$, set iteration $k$
    **output** $x_{best}$
    **initialize** $x \leftarrow x_0$, $T \leftarrow T_0$, $k \leftarrow 0$
    **while** $k \leq k_{max}$ **and** $T \geq T_f$ **do**
        $x_k \leftarrow$ NEIGHBOR($s$)
        $\Delta f \leftarrow f(x_k) - f(x)$
    **if** $\Delta f < 0$ **or** RANDOM(0, 1) $\leq P(\Delta f, T)$ **then**
            $x \leftarrow x_k$
    **end if**
        $T \leftarrow$ COOLING($T$, $k$, $k_{max}$)
        $k \leftarrow k + 1$
    **end while**
    $x_{best} \leftarrow x$
    **return** $x_{best}$

---

In essence, both GA and SA are heuristics. We employ these two tailored algorithms to validate the correctness of resolving the model and deriving a solution of a real case. In addition, with a reasonably acceptable gap of optimized results between the two algorithms, it turns out that a near-optimal solution is attained with a compatibility of efficiency. The next section practically demonstrates the modeling effectiveness and solution efficiency as per a real case of Shanghai shuttle transit.

## 5. Case Study

### 5.1. Data and Parameter Settings

The model established is implemented for a real case of shuttle buses that belong to the feeder system serving Jinshan Railway in Shanghai, China. The loop route of the shuttle bus creates a service circle that covers an area within a radius of 5 km. The *origin station* is the first-departure bus stop near Tinglin Railway Station. The working area and living area near the *origin station* are regarded as the demand districts. The shapes of these areas are depicted as a rectangle or a triangle, which facilitates the seeking of the destination centroid. As shown in Figure 6, demand communities are distributed along the east and west directions from the *origin station*.

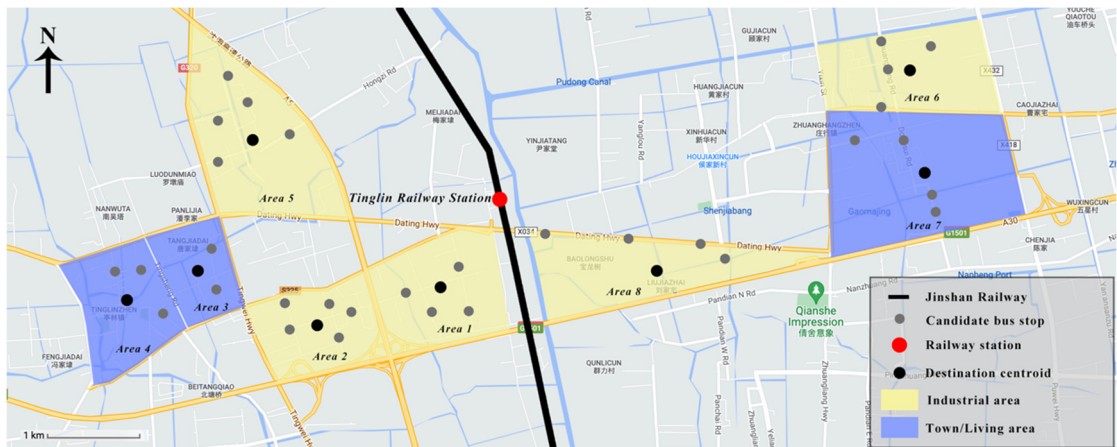

**Figure 6.** An example of a shuttle bus service in Shanghai, China.

The historical and officially released data serves as an input basic. The average headway of Jinshan Railway at peak hours is 20 min. Assume (hypothetically) that the delay time of a train will be 15 min. The train arrival time of each subprocess studied is shown in Table 3. Due to the disturbance, the headway in subprocess 2 becomes 35 min, and the headway in subprocess 3 becomes 5 min.

**Table 3.** The train arrival time and status of each subprocess.

| Status | Subprocess | Train Arrival Time |
|---|---|---|
| Normal | 1 | 7:30 |
| Abnormal | 2 | 7:50 |
| | 3 | 8:25 |
| Transition | 4 | 8:30 |
| Normal | 1 | 8:50 |
| Normal | 1 | 9:10 |

The arrival rate $\lambda$ of passengers is 26 pax/min based on practical, actual observation. Referring to Yu et al. [17], boarding and alighting time per passenger $\delta$ is set as 1.7 s. The fixed time of opening/closing door $\delta_0$ is 4.0 s. Values of parameters $c_o$, $c_{iv}$, and $c_{wv}$ that refer to Chen et al. [32] are valued as 2.5 USD/min, 0.17 USD/min, and 0.25 USD/min, respectively.

According to Assumption (2), the OD matrix of passengers is usually stable and identical. Thus, based on the practical survey and historical data, the number of passengers exiting from Tinglin Railway Station is 200 pax/train and the average running speed of the shuttle bus is 30 km/h. According to Constraint (10), the minimum headway $h_{min}$ and maximum headway $h_{max}$ are set to 1 and 30 min, respectively. Besides, the proportion of passengers arriving at each destination area can be obtained in Table 4. There are three gates at the exit. The vehicle capacity of the shuttle bus is 50 pax/vehicle.

**Table 4.** Percentage of arrival passengers in each destination area (100%).

| Index of Destination Area | Reach Ratio |
|---|---|
| 0 | 0 |
| 1 | 0.2 |
| 2 | 0.2 |
| 3 | 0.05 |
| 4 | 0.2 |
| 5 | 0.05 |
| 6 | 0.15 |
| 7 | 0.1 |
| 8 | 0.05 |

*5.2. Computation Results and Solution Quality*

The GA and SA used in this paper is programmed by Python 3.9 on a personal computer with Inter Core 8th i5 CPU @ 1.40 GHz and 8.00 GB RAM. In predetermining the initial parameters of GA from the experimental experience, it is reasonable to predefine the population size as 100, and the maximum number of iterations as 500, respectively. The probabilities of crossover and mutation are 0.9 and 0.001, respectively. For SA, it is reasonable to predefine starting and termination temperatures $T_0 = 100$, $T_f = 1 \times 10^{-9}$, and cooling rate as 150.

Two candidate stops are examined for serving each of the survey communities and an alternative one is elected by the proposed model. We predetermine a bus stop near the destination centroid within the communities so as to begin with the global iteration. Figures 7 and 8 exhibit the feasible links between stops to generate an optimized shuttle bus route by GA and SA, respectively. The same optimized routes are obtained by two different algorithms, whose elected route stops are indexed by 0-1-2-4-3-5-8-6-7-0. The length of the route is 15.44 km. Next, the available headway range is texted randomly in each subprocess to obtain the optimized result. Figures 9 and 10 illustrate that the fitness value decreases continuously along with algorithmic iteration, and the speed of convergence in SA is faster compared to GA. In both algorithms, the same results are obtained. We find the optimized solution in which the total cost is USD 9545 and the headway of each sub-process is 5, 7, 1,

5, and 5 min, respectively. Detailed results of GA and SA are shown in Table 5, where NIO refers to the number of iterations already when converging.

**Table 5.** Results of GA and SA.

| Algorithms | Optimized Routes | Headway of Each Sub-Process | Costs (USD) | CPU_Time (s) [1] | NIO [1] |
|---|---|---|---|---|---|
| GA | 0-1-2-4-3-5-8-6-7-0 | 5-7-1-5-5 | 9545 | 125.76 | 43 |
| SA | 0-1-2-4-3-5-8-6-7-0 | 5-7-1-5-5 | 9545 | 137.32 | 33 |

[1] CPU_time refers to the computation times when deriving a converging result; NIO is the number of iterations already when converging.

As a comparable metric, the item *no-planning* timetable refers to unchanged departure times and headway as the initial plan regardless of punctuality or disturbance undertaken. In Figure 11a, the headway of the *no-planning* timetable is regulated to be 5 min as a comparable metric whereas Figure 11b demonstrates the optimized headways obtained by GA which shift in each subprocess. The departure times of the *no-planning* timetable and adjusted timetable are shown in Table 6. Figure 12 shows the cost comparison of the two timetables. We find out that passengers in-vehicle travel time is unchanged when applying either of the two timetables. The optimized timetable requires more trips than the *no-planning* one, which causes a 10.2% increase to the operation cost compared to the cost of the *no-planning* timetable. On the other hand, in seeking the comparison of waiting time between them, a noteworthy reduction of 49.1% results from using the optimized timetable. The decrease is due to timely accommodation of the majority of congested accumulation passengers. To conclude, the optimized timetable results in a 7.6% reduction for the total objective compared to the *no planning* timetable.

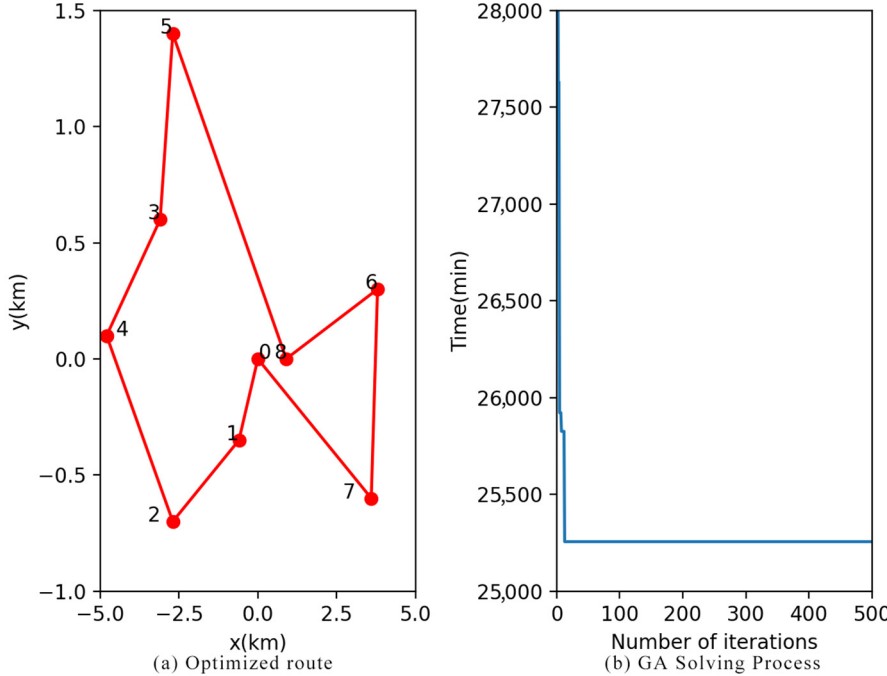

**Figure 7.** The optimized route for shuttle bus service by GA.

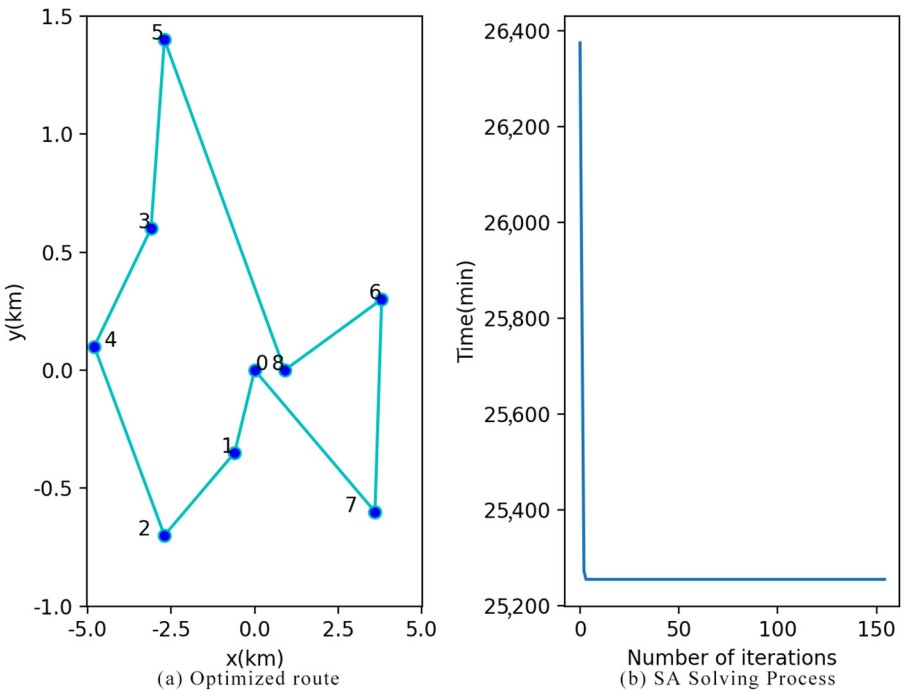

**Figure 8.** The optimized route for shuttle bus service by SA.

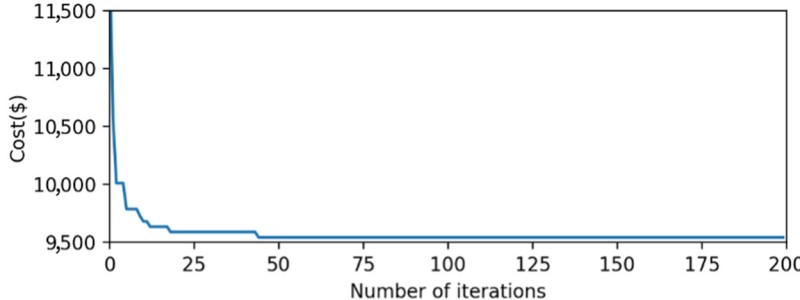

**Figure 9.** Obtaining the optimized timetable by GA.

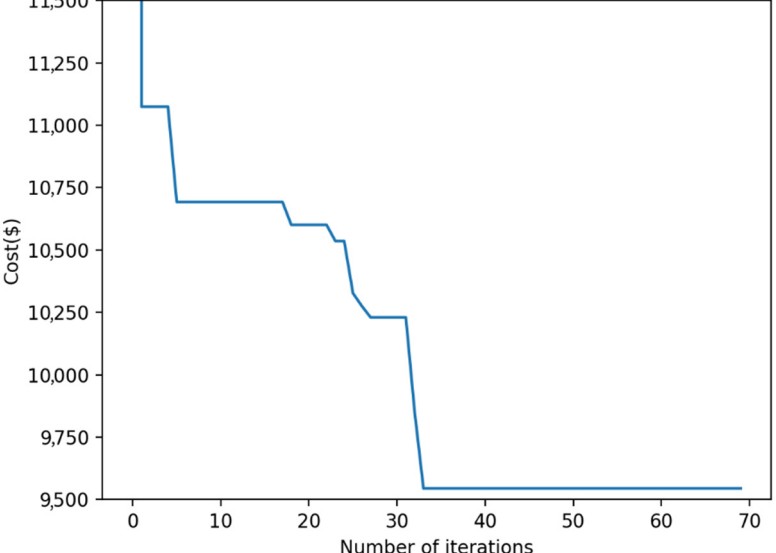

**Figure 10.** Obtaining the optimized timetable by SA.

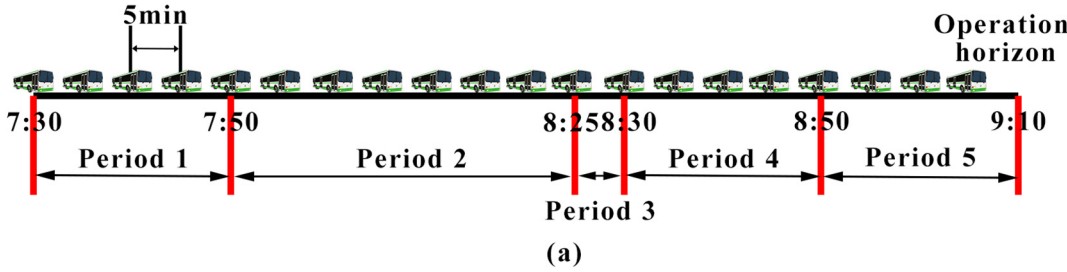

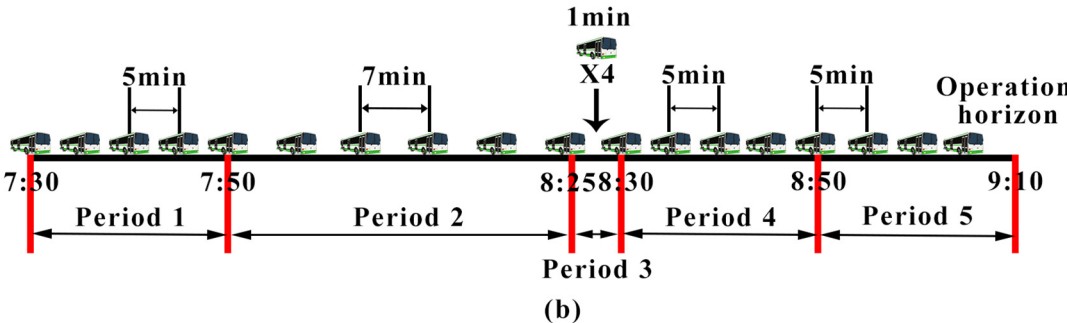

**Figure 11.** Optimized departure timetable in each subprocess: (**a**) *No planning* timetable; (**b**) optimized timetable.

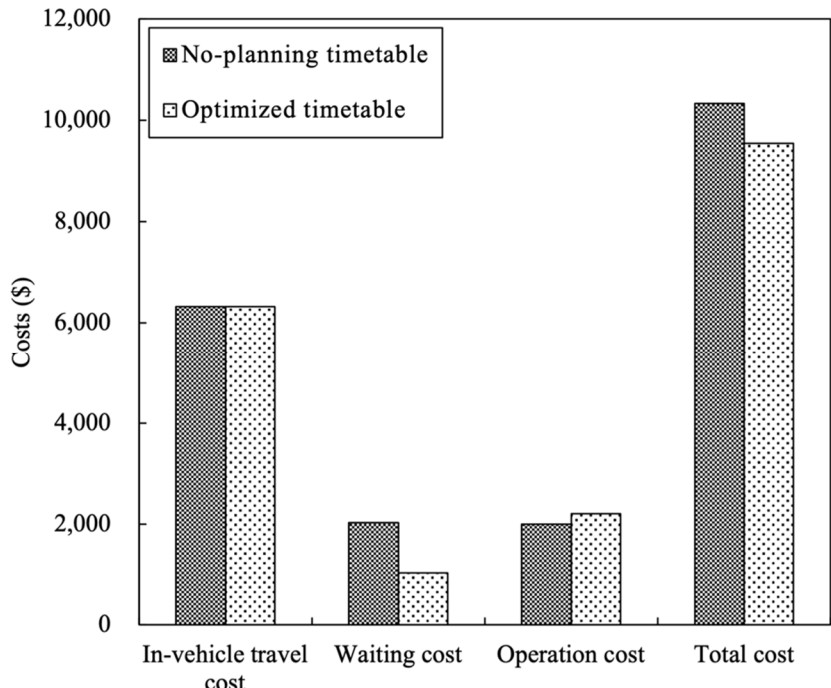

**Figure 12.** Cost comparison of the two plans.

**Table 6.** Detailed *no planning* and optimized bus timetable.

| No Planning | | | | | | Optimized | | | | | |
|---|---|---|---|---|---|---|---|---|---|---|---|
| Index | Trip | Departure Times | Index | Trip | Departure Times | Index | Trip | Departure Times | Index | Trip | Departure Times |
| 1 | $b_{11}$ | 7:30 | 12 | $b_{31}$ | 8:25 | 1 | $b_{11}$ | 7:30 | 12 | $b_{33}$ | 8:27 |
| 2 | $b_{12}$ | 7:35 | 13 | $b_{41}$ | 8:30 | 2 | $b_{12}$ | 7:35 | 13 | $b_{34}$ | 8:28 |
| 3 | $b_{13}$ | 7:40 | 14 | $b_{42}$ | 8:35 | 3 | $b_{13}$ | 7:40 | 14 | $b_{35}$ | 8:29 |
| 4 | $b_{14}$ | 7:45 | 15 | $b_{43}$ | 8:40 | 4 | $b_{14}$ | 7:45 | 15 | $b_{41}$ | 8:30 |
| 5 | $b_{21}$ | 7:50 | 16 | $b_{44}$ | 8:45 | 5 | $b_{21}$ | 7:50 | 16 | $b_{42}$ | 8:35 |
| 6 | $b_{22}$ | 7:55 | 17 | $b_{51}$ | 8:50 | 6 | $b_{22}$ | 7:57 | 17 | $b_{43}$ | 8:40 |
| 7 | $b_{23}$ | 8:00 | 18 | $b_{52}$ | 8:55 | 7 | $b_{23}$ | 8:04 | 18 | $b_{44}$ | 8:45 |
| 8 | $b_{24}$ | 8:05 | 19 | $b_{53}$ | 9:00 | 8 | $b_{24}$ | 8:11 | 19 | $b_{51}$ | 8:50 |
| 9 | $b_{25}$ | 8:10 | 20 | $b_{54}$ | 9:05 | 9 | $b_{25}$ | 8:18 | 20 | $b_{52}$ | 8:55 |
| 10 | $b_{26}$ | 8:15 | | | | 10 | $b_{31}$ | 8:25 | 21 | $b_{53}$ | 9:00 |
| 11 | $b_{27}$ | 8:20 | | | | 11 | $b_{32}$ | 8:26 | 22 | $b_{54}$ | 9:05 |

*5.3. Sensitivity Analysis*

Examining the variable change possibilities induces us to explore the quantifiable sensitivity of inputs, parameters, and constraints. In detail, the sensitivity analysis of the overall objective is performed with regard to different fluctuations on delay times (input), arrival rates measured by number of gates (parameter), and vehicle capacity (constraint).

5.3.1. Delay Times

In this section, we test a delay range between 1 min and 19 min which is derived from the commuter railway. In order to clarify the impact of different delay scales, a trend of fitness value is calculated.

Figure 13 primarily depicts the variation in total cost from USD 9450 to USD 9600 in response to delay time ranges from 1 min to 17 min. Notice that the delay time approaches the upper bounds of 18 min and 19 min, which results in a quickly increased cost of USD 9698 and USD 9845, respectively, and a markedly higher than the average cost. As investigated for the average cost of USD 9510 based on the 1st–17th min delay situations, the two heaviest delay scenarios, namely 18 and 19 min, cause a 1.98% and 3.53% increment of total cost, respectively. Yielding to the larger delays (such as 18–19 min), the swift growth on accumulated demand allows more trips (fleet sizes) to accommodate the layover passengers; however, these generate greatly increasing operation costs.

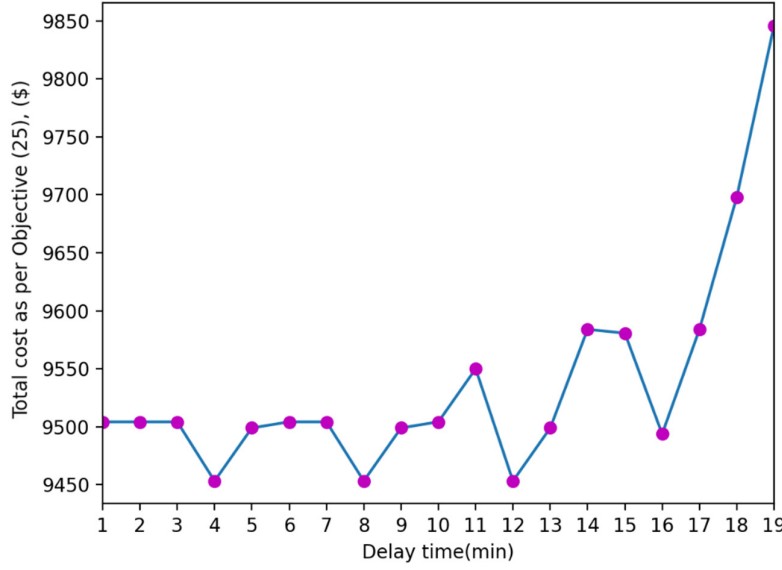

**Figure 13.** Sensitivity analysis for delay time.

### 5.3.2. Arrival Rates

Attention to infrastructure contributes to clarifying the efficiency of users exiting the railway. In practice, sites where passengers go through the gates are more inclined to produce huge crowds; it is the so-called bottleneck. Typically, we treat the number of gates as the core element of impact on passenger arrival rates. The observation of the number of gates in Figure 14 fulfills this test. There is a range from 1 to 10, which covers the common infrastructures in practice.

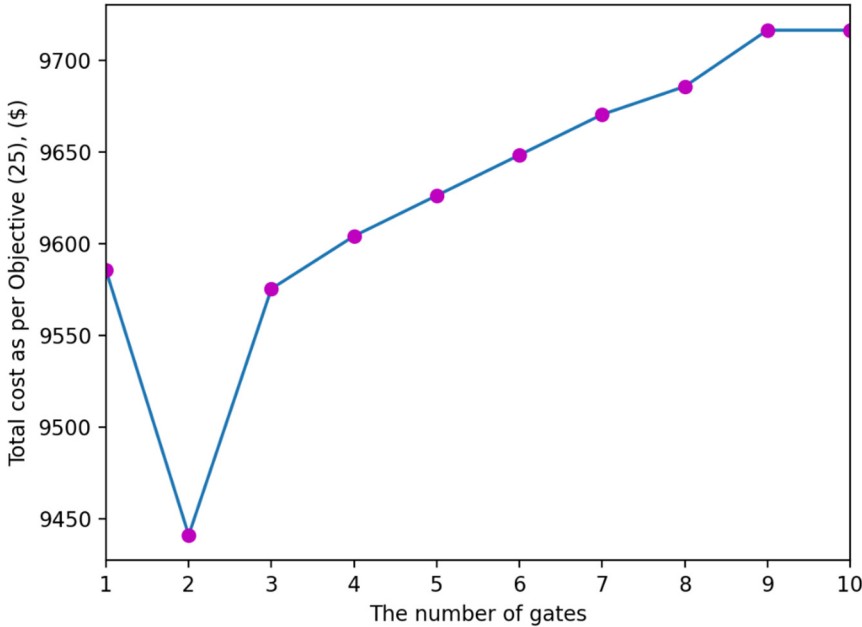

**Figure 14.** Sensitivity analysis for the number of gates.

A reduction (like a valley) of 1.51% and 1.40% in the value of two gates is represented in Figure 14 with a comparison to one gate and three gates, respectively. Clearly, their values are USD 9585, USD 9441, and USD 9575. Essentially, the number of gates possesses a positive correlation with passenger arrival rate. Yielding lesser bus capacity than train capacity, the frequency of the shuttle bus increases depending on the demand level measured by the number of gates passed. Until the number of gates reaches nine and ten or more, the total cost is stabilized at USD 9716. It suggests that, until the upper threshold is reached, the higher the passengers' arrival rate is, the longer the waiting times.

### 5.3.3. Vehicle Capacities

Last but not least, Figure 15 shows the investigation of the performance of multiple vehicle capacities. The capacity variable is tested to pursue the tradeoff between service frequency (as the reciprocal of headway) and waiting times as per a range of 15 to 120 pax. In the capacity range from 15 to 70 pax, lower capacity requires more trips/frequencies to serve, but it generates greater operation costs. On the other hand, along with the capacity increasing from 70 to 120 pax, one vehicle/trip can accommodate more waiting passengers, resulting in lower frequency. The longer headway poses drastically increased passenger waiting costs that account for the total cost and thereby heavily exceed 70 pax/veh. In particular, the total cost of USD 9433 with $C_b = 70$ indicates that the best compromise result emerging from the model is obtained by attaining a tradeoff between the minimal sum of operation and the waiting times, as demonstrated in Figure 15.

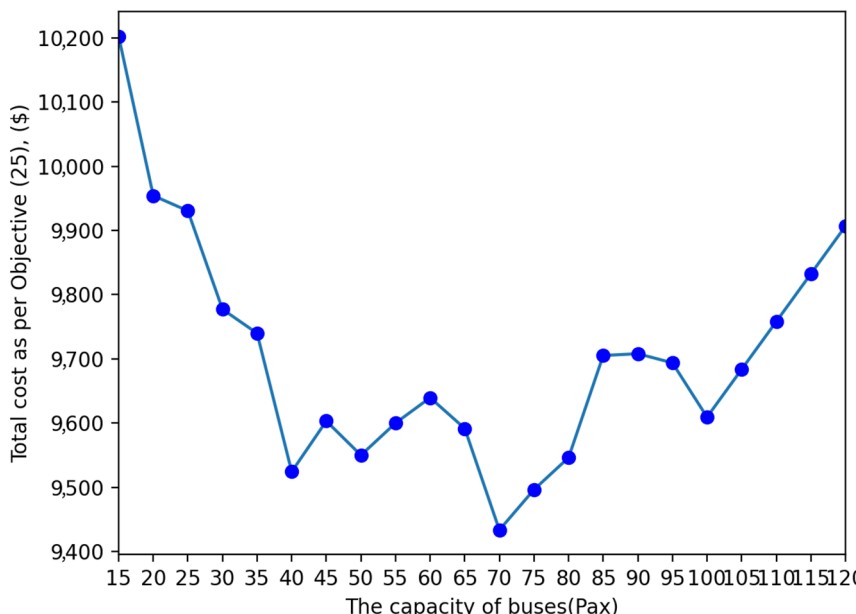

**Figure 15.** Sensitivity analysis for the vehicle capacity of buses.

## 6. Conclusions

This study focuses on shuttle bus timetable adjustment in response to commuter railway disturbances that specifically refer to behind-schedule delays. In an attempt to achieve last-mile convenience required by sustainable transportation, shuttle bus timetables are dedicated to offering synchronized connections in order to attain a more user-oriented service, favorable to transfer between both modes (i.e., from railway to bus). Thus, enhancing the capacity of the shuttle bus to handle unforeseen/fluctuated demand derived from behind-schedule railway delays is sufficient to warrant a valid model and solution; most previous studies paid less attention to the follow-up adjustment, even if the initial disturbance has already been addressed. To bridge this gap, not only are multiscale railway delays taken into consideration, but shuttle bus route and stop formulation is also facilitated. An integer programming model is proposed. The nature of the entire mathematical formulation of the model is non-linear, prompting us to seek decomposition and a heuristic method. The objective is to minimize passengers' waiting, travelling, and walking costs as well as the operation costs of the entire route yielding to the shortest path, in principle, between stops and centroids. Numerical results of the Shanghai case study validate 7.6% savings of the total cost by using our approach with a comparison between an adjustment timetable and the *no-planning* one. Moreover, they validate a 49.1% reduction to passengers' waiting times.

The threefold concentration of the sensitivity analysis on delay times, passenger arrival rates, and the upper bound of the capacity threshold are conducted to exploit the sophisticated nature of the modeling formulation. If we elaborate, this threefold concentration involves (i) the two heaviest delays (i.e., 18 and 19 min) leading a sharp growth of 1.98% and 3.53% because of the additional trips/vehicles required for accumulated demand in the delay horizon; (ii) the test of passengers' arrival rate, mainly impacted by the number of gates, which finds a valley (USD 9441) floor and steady peak value (USD 9716); (iii) optimal capacity value (70 pax) sought along with its bilateral variation of capacity, which presents a compatible optimum for the operation and passenger waiting costs.

This study, aimed at striking a successor solution to the adjustment problem of shuttle bus response to a behind-schedule predecessor railway, elucidates the potential of sustaining future research. As successive solutions narrow the focus of future research, our attention is directed to the following aspects requiring improvement or solutions:

1. The disruptions on the commuter railway, which spurs the adjustment of vehicle scheduling and renders crew scheduling infeasible. Under this circumstance, the knock-on effect of spreading the delay should be taken into consideration;
2. Traffic conditions should be considered in terms of the randomness of traveling time;
3. Passengers' boarding and alighting times tend to be precisely tracked, and this needs to be improved specifically in relation to the degree of in-vehicle crowding.

**Author Contributions:** Conceptualization, Y.F., Z.C. and S.Z.; methodology, Y.F. and Z.C.; software, Y.F.; validation, Y.F.; formal analysis, Y.F. and Z.C.; investigation, Y.F.; resources, Y.F.; data curation, Y.F.; writing—original draft preparation, Y.F.; writing—review and editing, Z.C.; visualization, Y.F.; supervision, Z.C. and S.Z.; project administration, Z.C. and S.Z.; funding acquisition, Z.C. and S.Z. All authors have read and agreed to the published version of the manuscript.

**Funding:** This study was supported by National Natural Science Foundation of China (72101127) and (72101126), by the Science and Technology Project of Jiangsu Province, China (BK20200978), by Graduate Research and Practice Innovation Program of School of Transportation and Civil Engineering, Nantong University, China (NTUJTXYGI2203), and sponsored by CCF-Tencent Open Fund (CCF-Tencent IAGR20220111).

**Institutional Review Board Statement:** Not applicable.

**Data Availability Statement:** Not applicable.

**Conflicts of Interest:** The authors declare no conflict of interest.

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
