# Peer review of "Shuttle Bus Timetable Adjustment in Response to Behind-Schedule Commuter Railway Disturbance"

_sustainability, doi:10.3390/su142416708_

Round 1

Reviewer 1 Report

This paper describes is to  a nonlinear integer programming model, in which the objective function is based on the waiting, travelling, and walking cost to passengers as well as operating cost of the route chosen. Vehicle capacity constraint and precise passenger waiting times are considered. Genetic algorithm and simulated annealing algorithm combined with a priori decomposition are employed to facilitate the solution efficiency. The article has an applied character. Mathematical models are described. The choice of the proposed methods is justified. The work of this paper has practical application.

1. Consider using CPM ("Critical Path Method") or PERT.

2. Research of efficiency of methods of Artificial Intelligence.

Author Response

Response to reviewers - sustainability-2081144, Dec.-2022

Shuttle bus timetable adjustment in response to behind-schedule commuter railway disturbance

Note: All changes and revisions made in the revised manuscript are highlighted for convenience.

Reviewer 1:

This paper describes is to a nonlinear integer programming model, in which the objective function is based on the waiting, travelling, and walking cost to passengers as well as operating cost of the route chosen. Vehicle capacity constraint and precise passenger waiting times are considered. Genetic algorithm and simulated annealing algorithm combined with a priori decomposition are employed to facilitate the solution efficiency. The article has an applied character. Mathematical models are described. The choice of the proposed methods is justified. The work of this paper has practical application.

Response: Thanks. We appreciate your view. 

  1. Consider using CPM ("Critical Path Method") or PERT.

Response: We think it is a very good innovation point and will use these two methods in future research. Thanks.

  1. Research of efficiency of methods of Artificial Intelligence.

Response: In Table 5 on P19, L528-L530, we studied and compared the efficiency of GA and SA, namely CPU_time and NIO. Thanks again.

Thank you for your reviewing effort!

Reviewer 2 Report

This article describes a method that optimizes shuttle bus routes and timetables  following a railway disturbance.

After an introduction where they describe the need for optimized shuttle bus routes and timetables, analyse existing research on shuttle bus optimization and explain their contributions, the authors describe the formal problem they aim to solve in parts 2 and 3, and explain their method to solve this problem in part 4. Part 5 is dedicated to a case study, where the athors show how on a given use case their problem formulation gives different (and more efficient) solutions compared to an approach that does not take users' waiting time into account. They then analyse the advantages of their approach and give paths to improve it in a conclusion.

The authors dedicate a significant portion of the article to the description of the parameters and variables of their problem, and to the formulation of the problem itself. This description is done in a formal way (with equations), but the authors also provide textual explanations to justify their formulation, and this part, which can be difficult to understand in other articles, is clear for the most part. The authors give less details on their methods to solve the formal problem ; I presume they mostly used known tools for that. The case study illustrates the interest of the method very didactically ; it shows very clearly that taking users' waiting time into account leads to more efficient solutions (from a global point of view), even if these solutions entail more trips and thus more costs to operate. Overall, the article is clear, the authors' chosen method is well described and sufficiently new, and the case study clearly shows its interest.

Apart from my reservations about the English language, I have a few remarks and questions, mainly about the formulation of the problem and the methods used to solve it.

Formulation of the problem:

Maybe it is clear for specialists of the domain, but the authors should specify that quantities w_{ij} and \delta_{i} represent totals over all the trips (and are not values for a single trip) [If I understand the model correctly]. If it is indeed the case, I have reservations about equation (13), because then the constant \delta_{0} should be multiplied by d_b, i.e. the number of trips that drop passengers at bus stop i (More generally, all terms of equation (1) have an element that takes into account how data are integrated over time (sum on U, or factors like d_{b}, d_{g}, w_{ij}, \delta_{i}), but in the equaltion for \delta_{i}, this ‘integrating factor’ is missing for the term with \delta_{0}).

I wonder if it should be specified that a passenger waiting time is necessarily nonnegative (i.e. wt_{u} \geq 0). With this formulation, is there something preventing negative waiting times (i.e. a user boarding a bus before arriving at the origin station)?

Solution algorithms

Fot the genetic algorithm:

l.345 stipulates that the optimal shuttle bus headway h’_{bs} is derived from the optimized route r’, which leads the reader to think that it is a straightforward step. Then, the flowchart in Figure 3 precises that the optimized headway is determined after another T iterations of the genetic algorithm. If it is the case, the formulation of l.345 should be modified to give this precision in the text (and not only in Figure 3).

In general:

In the description of the genetic algorithm and of the simulated annealing algorithm, no detail is given on the crossover tool and the mutation operations (for the genetic algorithm), or on how the neighborhood of a solution is determined (for the SA algorithm), but these elements are crucial for the understanding and the evaluation of these algorithms. If the authors do not develop a novel approach for these, they could either briefly describe how these operations are done, or refer to other articles that describe them if they use well-known transformations to achieve crossover or mutation.

IAt l.298,  Objective function (40) should be Objective function (36)

In  their algorithms, the authors first compute their solution route r', and then compute optimized timetables with this route. If I understand correctly l.298 to 305, the optimization of the route is done by solving the smaller routing problem (as opposed to the complete routing-timetabling problem), for which the objective function is given by equation (36), with the objective to minimize the cost (in time) of a single trip, with a fixed proportion of passengers, without taking timetabling issues and associated costs into account. If it is indeed the case, then the description of the genetic algorithm and of the SA algorithm are confusing, because the reader is lead to think that the initialization, and then the different stages of one iteration, are done for the whole problem, whereas the authors use the same algorithm twice with very different solution spaces. Apart from the basic structure of the genetic algorithm, the two uses of the genetic algorithm have very different initialization, evaluation, selection, crossover and mutation stages, and the authors need to give two more detailed decriptions of the genetic algorithm, the first one for the routing problem and the second one for the timetabling problem.

The English language is generally understandable, but there are many grammar mistakes, awkward sentence constructions, and , less frequently, confusing sentences where grammar and choice of word make the intent of the authors ambiguous. As a result, a thorough editing of English language is required.

Here are some editing suggestions for the abstract only (the remainder of the article would to be checked with the same meticulousness:

l.8 : the unscheduled issues : it is difficult to understand what is meant here. Do the authors mean unplanned problems, or a more specific issues related to train schedules ?

l.9 : the sustainable transportation → sustainable transportation

l.9 : facilitates → I would rephrase as ‘The study describes tools that enable / facilitate shuttle bus timetabling adjustment

l.9 : timetabling adjustment : I would rather write timetable adjustment to name this concept

l.11 : is divided from four stages → is divided in four stages

l.11 : allowing for the multiple delay extends → allowing for different delay ranges

l.11-12 : the focus of the problem and its solution relates jointly to… : this formulation is confusing ; I would rewrite (for example) ‘The problem and its solution involve different elements, such as…

l.14 : […] and walking cost to passengers → […] and walking costs for passengers

l.14 : as well as operation cost → as well as the operation cost

l.15 : Vehicle capacity constraint… → Vehicle capacity constraints… (or The constraints on vehicle capacity…)

l.16 : Genetic algorithm and simulated annealing algorithm […] are employed… → A genetic algorithm and a simulated algorithm […] are used…

l.17 : to facilitate the solution efficiency → to compute an efficient solution

Author Response

Response to reviewers - sustainability-2081144, Dec.-2022

Shuttle bus timetable adjustment in response to behind-schedule commuter railway disturbance

Note: All changes and revisions made in the revised manuscript are highlighted for convenience.

Reviewer 2:

This article describes a method that optimizes shuttle bus routes and timetables  following a railway disturbance.

After an introduction where they describe the need for optimized shuttle bus routes and timetables, analyse existing research on shuttle bus optimization and explain their contributions, the authors describe the formal problem they aim to solve in parts 2 and 3, and explain their method to solve this problem in part 4. Part 5 is dedicated to a case study, where the authors show how on a given use case their problem formulation gives different (and more efficient) solutions compared to an approach that does not take users' waiting time into account. They then analyse the advantages of their approach and give paths to improve it in a conclusion.

The authors dedicate a significant portion of the article to the description of the parameters and variables of their problem, and to the formulation of the problem itself. This description is done in a formal way (with equations), but the authors also provide textual explanations to justify their formulation, and this part, which can be difficult to understand in other articles, is clear for the most part. The authors give less details on their methods to solve the formal problem ; I presume they mostly used known tools for that. The case study illustrates the interest of the method very didactically ; it shows very clearly that taking users' waiting time into account leads to more efficient solutions (from a global point of view), even if these solutions entail more trips and thus more costs to operate. Overall, the article is clear, the authors' chosen method is well described and sufficiently new, and the case study clearly shows its interest.

Apart from my reservations about the English language, I have a few remarks and questions, mainly about the formulation of the problem and the methods used to solve it.

Response: Thanks for your comments. We revised the expression of some variables and ensured all constraints to explain more clearly. In addition, we explained the algorithm used in the paper more explicitly. Indeed, the language of the paper had been polished and ensured before submitted. We can prove it. We provided an attach as a supplement. Of Couse, your advice and reminder are appreciated much, which motivate us check and improve the whole paper.

  1. Maybe it is clear for specialists of the domain, but the authors should specify that quantities w_{ij} and \delta_{i} represent totals over all the trips (and are not values for a single trip) [If I understand the model correctly]. If it is indeed the case, I have reservations about equation (13), because then the constant \delta_{0} should be multiplied by d_b, i.e. the number of trips that drop passengers at bus stop i (More generally, all terms of equation (1) have an element that takes into account how data are integrated over time (sum on U, or factors like d_{b}, d_{g}, w_{ij}, \delta_{i}), but in the equaltion for \delta_{i}, this ‘integrating factor’ is missing for the term with \delta_{0}).

 Response: We explained  and  more clearly in Table 2 on P8 (highlighted), which are based on all the trips. In addition, the constant  does need to be multiplied by , and we re-modified Constraint (13) (now that is Constraint (14)) on P9 (highlighted). Meanwhile, we modified part of the interpretation of Constraint (13) (now that is Constraint (14)) on P10, L282-L284 (highlighted).

  1. I wonder if it should be specified that a passenger waiting time is necessarily nonnegative (i.e. wt_{u} \geq 0). With this formulation, is there something preventing negative waiting times (i.e. a user boarding a bus before arriving at the origin station)?

 Response: Thanks for your good question. We avoided negative waiting time by adding Constraint (13) on P9 (highlighted) and Constraint (34) on P11 (highlighted). At the same time, we added (and highlighted) the explanation of Constraint (13) on P10, L280-L282.

  1. l.345 stipulates that the optimal shuttle bus headway h’_{bs} is derived from the optimized route r’, which leads the reader to think that it is a straightforward step. Then, the flowchart in Figure 3 precises that the optimized headway is determined after another T iterations of the genetic algorithm. If it is the case, the formulation of l.345 should be modified to give this precision in the text (and not only in Figure 3).

 Response: Good point. In order to more clearly describe the application of GA in this paper, we subdivided Section 4.2 into Section 4.2.1 (P11-P12, L319-L362, highlighted) and Section 4.2.2 (P13, L366-L382, highlighted). These two sub-sections introduced how to use GA to achieve the optimized routes and timetable, and introduced their input and output in detail (P12, L361-L362; P13, L379-L380, highlighted), respectively.

  1. In the description of the genetic algorithm and of the simulated annealing algorithm, no detail is given on the crossover tool and the mutation operations (for the genetic algorithm), or on how the neighborhood of a solution is determined (for the SA algorithm), but these elements are crucial for the understanding and the evaluation of these algorithms. If the authors do not develop a novel approach for these, they could either briefly describe how these operations are done, or refer to other articles that describe them if they use well-known transformations to achieve crossover or mutation.

 Response: In order to better illustrate the crossover and mutation operation process of GA, we added Fig.3 on P12, L349-L350, Fig.4 on P12, L356-L357, and the corresponding explanation on P12, L342-L347 and L353-L355 (highlighted). In addition, we added an explanation for the neighborhood selection of SA in Section 4.3.1 on P14-P15, L413-L415 (highlighted).

  1. IAt l.298, Objective function (40) should be Objective function (36)

 Response: Good pinpoint. We modified the corresponding number on P11, L303.

  1. In their algorithms, the authors first compute their solution route r', and then compute optimized timetables with this route. If I understand correctly l.298 to 305, the optimization of the route is done by solving the smaller routing problem (as opposed to the complete routing-timetabling problem), for which the objective function is given by equation (36), with the objective to minimize the cost (in time) of a single trip, with a fixed proportion of passengers, without taking timetabling issues and associated costs into account. If it is indeed the case, then the description of the genetic algorithm and of the SA algorithm are confusing, because the reader is lead to think that the initialization, and then the different stages of one iteration, are done for the whole problem, whereas the authors use the same algorithm twice with very different solution spaces. Apart from the basic structure of the genetic algorithm, the two uses of the genetic algorithm have very different initialization, evaluation, selection, crossover and mutation stages, and the authors need to give two more detailed decriptions of the genetic algorithm, the first one for the routing problem and the second one for the timetabling problem.

 Response: Good point. To solve this problem, we subdivided Section 4.2 into Section 4.2.1 (P11-P12, L319-L362, highlighted) and Section 4.2.2 (P13, L366-L382, highlighted), and Section 4.3 into Section 4.3.1 (P14-P15, L407-L420, highlighted) and Section 4.3.2 (P15, L421-L430, highlighted), which describe the application of GA and SA in route and timetable optimization, respectively. In each sub-section, their process was described in detail. On P11, L323-L324; P13, L371-L373; P14, L409-410; P15, L424-427, the expression of the initial population was added (highlighted). On P11, L327-L328; P13, L375-L377, the expressions of the personal evaluation were added (highlighted).

  1. The English language is generally understandable, but there are many grammar mistakes, awkward sentence constructions, and , less frequently, confusing sentences where grammar and choice of word make the intent of the authors ambiguous. As a result, a thorough editing of English language is required.

Here are some editing suggestions for the abstract only (the remainder of the article would to be checked with the same meticulousness:

l.8 : the unscheduled issues : it is difficult to understand what is meant here. Do the authors mean unplanned problems, or a more specific issues related to train schedules ?

 Response: Corrected and highlighted on P1, L8. Thanks.

  1. l.9 : the sustainable transportation → sustainable transportation

 Response: Corrected on P1, L9.

  1. l.9 : facilitates → I would rephrase as ‘The study describes tools that enable / facilitate shuttle bus timetabling adjustment

 Response: Corrected on P1, L9.

  1. l.9 : timetabling adjustment : I would rather write timetable adjustment to name this concept

 Response: Corrected on P1, L2, L9, L13, L21 and L42; P2, L51 and L59.

  1. l.11 : is divided from four stages → is divided in four stages

 Response: Corrected on P1, L11.

  1. l.11 : allowing for the multiple delay extends → allowing for different delay ranges

 Response: Corrected on P1, L11; P2, L51-L52 and L60.

  1. l.11-12 : the focus of the problem and its solution relates jointly to… : this formulation is confusing ; I would rewrite (for example) ‘The problem and its solution involve different elements, such as…

 Response: Corrected on P1, L11-L12.

  1. l.14 : […] and walking cost to passengers → […] and walking costs for passengers

 Response: Corrected on P1, L14.

  1. l.14 : as well as operation cost → as well as the operation cost

 Response: Corrected on P1, L14.

  1. l.15 : Vehicle capacity constraint… → Vehicle capacity constraints… (or The constraints on vehicle capacity…)

 Response: Corrected on P1, L15.

  1. l.16 : Genetic algorithm and simulated annealing algorithm […] are employed… → A genetic algorithm and a simulated algorithm […] are used…

 Response: Corrected on P1, L16.

  1. l.17 : to facilitate the solution efficiency → to compute an efficient solution

 Response: Corrected on P1, L17. Thanks for all language suggestions.

Thank you for your reviewing effort!